# Antimicrobial Lock Therapy in Clinical Practice: A Scoping Review Protocol

**DOI:** 10.3390/mps3010016

**Published:** 2020-02-12

**Authors:** Aniello Alfieri, Sveva Di Franco, Maria Beatrice Passavanti, Maria Caterina Pace, Agata Stanga, Vittorio Simeon, Paolo Chiodini, Sebastiano Leone, Vettakkara Kandy Muhammed Niyas, Marco Fiore

**Affiliations:** 1Department of Women, Child and General and Specialized Surgery, University of Campania “Luigi Vanvitelli”, 80138 Naples, Italy; anielloalfieri@gmail.com (A.A.); beatrice.passavanti@libero.it (M.B.P.); caterina.pace@libero.it (M.C.P.); aghy89@libero.it (A.S.); 2Department of Public, Clinical and Preventive Medicine, Medical Statistics Unit, University of Campania Luigi Vanvitelli, 80138 Naples, Italy; vittoriosimeon@gmail.com (V.S.); paolo.chiodini@gmail.com (P.C.); 3Division of Infectious Diseases, Department of Internal Medicine, San Giuseppe Moscati Hospital, Contrada Amoretta, 83100 Avellino, Italy; sebastianoleone@yahoo.it; 4Department of Medicine, All India Institute of Medical Sciences, New Delhi 110029, India; niyas987@gmail.com

**Keywords:** antimicrobial lock therapy, ALT, anti-infective agents, bacteremia, biofilm, central venous catheters, catheter-related bloodstream infection, central-line-associated bloodstream infection, scoping review

## Abstract

Our objective is to review the scientific literature on the use of antimicrobial lock therapy (ALT). To achieve this result, our scoping review will address the following seven key questions: (1) Who are the patients who will benefit from this technique? (2) What are the techniques utilized? (3) What are the settings in which the technique is performed? (4) When the technique is performed? (5) Why the technique is performed? (6) How the technique is performed? (7) In how much amount, of such technique performed? This review considers all studies published in full and in peer-reviewed journals, with no restrictions on language, on the year of publication and age of the participants. Both randomized controlled trials and observational studies will be included. This scoping review has been planned on a five-stage framework: 1. Identifying the review question; 2. identifying relevant studies; 3. study selection; 4. charting the data; 5. collating, summarizing, and reporting the results. It is conducted in accordance with the Preferred Reporting Items for Systematic Reviews and Meta-Analyses Guidelines. The databases utilized will include MEDLINE via PubMed, EMBASE and Cochrane Central Register of Controlled Trials and Grey Literature. SCOPING REVIEW REGISTRATION: Open Science Framework https://osf.io/vphwm/.

## 1. Introduction

About 250,000 bloodstream infections are diagnosed each year, and most are related to the presence of intravascular devices [1]. Central line-associated bloodstream infections (CLABSI) are defined as a laboratory-confirmed bloodstream infection that recognize a central line as the focus of infection, typically occurring > 48 h after central line placement [2]. It has been estimated that the CLABSI rate in the United States (U.S.) intensive care units (ICU) is 0.8 per 1000 central line days [2]. The ICU surveillance data on 703 intensive care units in 50 countries, furnished by the International Nosocomial Infection Control Consortium (INICC) between January 2010 and December 2015, reported a CLABSIs rate of 4.1 per 1000 central line days [3]. The higher rate of CLABSIs in Latin America, Europe, Eastern Mediterranean, Southeast Asia, and Western Pacific, if compared to the incidence in the United States, seems to be due to insufficient resources and experience in infection control of the developing countries [4]. It has to be noticed that compared to other healthcare-associated infections (ventilator-associated pneumonia, surgical site infections, Clostridium difficile infection, and catheter-associated urinary tract infections), CLABSIs are the costliest to manage, accounting for about $46,000 per case in the United States. 2. CLABSIs are also among the most preventable health care infections. Institution of proper catheter care measures reduces CLABSI rate by 65–70% [5]. Proper catheter care includes following aseptic techniques during insertion, identifying the best insertion site with a preference for subclavian vein insertion, surveillance measures, and management strategies. These prophylactic measures, described above, should have to lead to a relevant decrement of CLABSIs incidence. However, these complications are still not completely eliminated [6,7,8]. Antimicrobial lock therapy (ALT) has been investigated in the prevention and treatment of central venous catheters-related infections. ALT consists, first, in the manual injection (“flush”) of a solution, generally normal saline, with the purpose of cleaning the inner lumen of the catheter, removing remnants of infused substances and maintaining its patency and, second in the filling of the catheter (“lock”) with a limited volume of the chosen lock solution in the intervals of time when the catheter is not in use, with the purpose of preventing lumen occlusion and/or bacterial colonization. G.A. Goossens in 2015, has clearly defined modalities of flushing and locking to take into account in our daily clinical practice [9]. Catheter lock technique would have been the more appropriate term for this technique as it would include both antibiotic and non-antibiotic lock solutions. In the current terminology ALT refers to all the techniques regardless of the lock solution used. Over the past few years, several randomized trials have investigated ALT and yielded promising results [10,11]. However, there are concerns regarding the emergence of antibiotic-resistant organisms, noninfectious complications, and the inability of previous studies to prove the additive benefit of lock solutions in conjunction with catheter care bundles [12,13,14]. All these factors delayed the inclusion of ALT into guidelines for the management of CLABSI until 2009 [15]. These guidelines in the section for General Management of Catheter-Related Infection at the point number 30 suggest that ALT should be used for catheter salvage (B-II); however, if antibiotic lock therapy cannot be used in this situation, systemic antibiotics should be administered through the colonized catheter (C-III) [15]. Hospitals have different institutional policies regarding CLABSI. According to the ALT Guideline at Stanford Hospital, this technique is highly recommended [16]. The Infectious Diseases Society of America (IDSA) guidelines for the diagnosis and management of CLABSI recommend ALT as adjunctive therapy specifically for catheter salvage in cases where the catheter is not removed [15]. Furthermore, because of the efficacy of the ALT in pediatric patients, there are several pediatric hospitals that are developing and implementing ALT guidelines [17]. The ESMO guidelines (2015) [18] and the recent literature [19], provide for the use of ALT in prevention for long-term infections associated with central venous catheters (CLABSI) in cancer patients. Even if several evidences support the use of ALT in seriously ill patients such as patients in need of palliative care, the effectiveness in patients in need of hemodialysis is still unclear [20]. Despite the elaboration of all these clinical practice protocols or guidelines, the most appropriate lock solution for central venous access devices is still to be defined [19]. Conceptually the ALT should be performed with a solution that a) has a low toxicity, b) cost-effective, c) has no risk of development of antibiotic microbial-resistance, d) has broad spectrum of activity, e) has no interaction with anticoagulant with a f) stability of antimicrobial activity over a prolonged period ( 24–48 h). There are several recent studies that tried to define the best lock solution to be used. These studies are different from each other according to their methodology and setting ranging from the network meta-analysis by Dang et al. published in 2019, to experimental models such as that proposed by Basas J. and colleagues. According to Dang et al. minocycline-ethylenediaminetetraacetic acid (EDTA) is effective in the prevention of CRBSI and exit-site infection [21], while the study by Basas J. et al. on rabbit model showed that liposomal amphotericin B (LAmB) ALT or anidulafungin ALT can be used to treat CLABSIs infections by Candida glabrata and Candida albicans [22]. Because of the wide range of available evidence on CLABSIs management, this scoping review unlike the previous systematic reviews [23] does not aim to give recommendations on the use of ALT but aims to furnish a map and a synthesis on this topic, offering a base for the development of new clinical practice indications and health policies. This scoping review aims to summarize the main indications for ALT and list the lock therapy solutions described in literature (not only antibiotic solutions). The systematic and validated review approach avoids the risk of generating conclusions based on studies of debatable quality.

## 2. Protocol Design

The scoping review is a study methodology for synthesizing primary research evidence. According to Grant and Booth [24], scoping reviews are "preliminary assessment of potential size and scope of available research literature.” Such reviews aim to identify nature and extent of research evidence. Its objective is to map the existing literature in an area of interest in terms of the size, type, and characteristics of the primary research. This scoping review began with the collaboration of a research team composed of experts in critical care medicine, infectious diseases, epidemiology, and research synthesis. It has been planned on the five stages framework proposed by Arksey and O’Malley [25] which has been further developed by Levac et al. [26] and the Joanna Briggs Institute [27]: 1. Identifying the review question; 2. identifying relevant studies; 3. study selection; 4. charting the data; 5. collating, summarizing, and reporting the results. Furthermore, it will be conducted in accordance with Preferred Reporting Items for Systematic Reviews and Meta-Analyses (PRISMA-ScR) guidelines [28]. The protocol was registered prospectively with the Open Science Framework on 13 December 2019 [29].

## 3. Patient and Public Involvement

As proposed by the original framework by Arksey and O’Malley [25], reference workout with key stakeholders (study groups of the societies of anesthesia and resuscitation, infectious diseases and palliative care along with patients and patients’ family organizations), to detect supplementary references around possible studies to be included, has been planned.

## 4. Stage 1: Identifying the Research Question

The scope of this review is to map the available literature on ALT. To achieve this result, our review will answer the following research questions:
Who are the patients who will benefit from this technique? (e.g., oncological patients, end-of-life patients, etc.)?What are the techniques utilized (e.g., chemical, antibiotic solution, etc.)?What are the settings in which the technique is performed? (e.g., intensive care unit, long-term care facilities, etc.)?When the technique is performed (e.g., first infection episode, recurrence of infection, etc.)?Why the technique is performed (e.g., prevention, treatment, etc.)?How the technique is performed (e.g., type of antibiotic or physical agent, etc.)?How much each technique is performed (substance concentration, therapy sessions per day, etc.)?

## 5. Stage 2: Identifying Relevant Studies

### 5.1. Eligibility Criteria

This scoping review will consider all studies (both controlled trials and observational studies, including controlled and uncontrolled studies) focusing on ALT, excluding experimental models. The studies must be published in full, in peer-review journals. The grey literature will also be consulted to find unpublished studies that could be included in our scoping review. There are no limitations regarding the year of publication, language, and age of the participants.

### 5.2. Search Strategy

A three-step search strategy was followed to find and analyze the studies published in peer-reviewed journals. A first limited search on MEDLINE, CINAHL was undertaken. This search was followed by the analysis of the text words contained in the title and abstract, and of the index terms used to describe articles. Once the information to develop a complete search strategy is obtained from previous sources, it is tailored for each database. The databases utilized include MEDLINE, EMBASE, and Cochrane Central Register of Controlled Trials (CENTRAL). A proposed search strategy for PubMed is detailed in Table 1. A subsequent search using keywords and index terms was then started through all included databases (see below for the list of databases). Several databases were consulted to examine the gray literature: Conference Proceedings Citation Index via Web of Science, Dissertation Abstracts via ProQuest, Google Scholar, Networked Digital Library of Theses and Dissertations, OpenDOAR and Open Grey. The reference list of all identified reports and articles is searched for additional studies; this includes the use of backward and forward citation tracking. Already published reviews references were used as a source for any study not found in the databases research. A final search used the reference list of all systematic reviews to identify the additional eligible studies.

## 6. Stage 3: Study Selection

The screening was performed in two phases, namely initial screening based on title and abstract, followed by a full-text screening of the eligible articles for final inclusion. The citation management software that was used to manage the selection process is Endnote VX9 (Clarivate Analytics, PA, USA) and duplicates were removed. Two authors (AA and SDF) independently evaluated the titles and abstracts of potentially eligible studies. Any disagreement on study eligibility between AA and SDF was resolved according to a discussion with a third reviewer (MF). The authors of the primary studies to be screened were contacted in the case of missing data.

## 7. Stage 4: Data Charting

Data were extracted independently using the Joanna Briggs Institute’s System (JBI) Data Extraction Form for Experimental/Observational Studies and the results were cross-checked. Any disagreements on study eligibility or data extraction were resolved according to a third reviewer’s opinion (MF). Studies considered as potentially relevant were analyzed in full and their reference list was imported into the JBI System for the Unified Management, Assessment and Review of Information (JBI SUMARI; The Joanna Briggs Institute, Adelaide, Australia) [27]. The selected citations’ full text was assessed in detail against the inclusion criteria by two independent reviewers. Reasons for exclusion of full-text studies not meeting the inclusion criteria were recorded and reported in this systematic review. Any conflict rising between the reviewers at each stage of the study selection process was resolved through debate, or by a discussion with a third reviewer (MF). The results of each step of the planned search was reported in detail in the closing report and presented in a Preferred Reporting Items for Systematic Reviews and Meta-analyses (PRISMA) flow diagram [28]. The bibliography of the included studies was screened and/or hand-searching of journals was performed. Citations of eligible studies retrieved in full were imported into JBI SUMARI [27]. The reasons for excluding studies on full text were provided in an appendix in the review.

## 8. Stage 5: Collating, Summarizing and Reporting the Results

The data extraction process is referred to as “charting the results,” providing a descriptive and logical summary of the observational studies retrieved. The JBI template study details, characteristics, and results extraction instrument are used to record the critical information of the source, such as: author(s), year of publication, origin/country of origin (where the study was published or conducted), aims/purpose, study population and sample size, methodology/methods, intervention type, comparator and details of these (e.g., duration of the intervention), duration of the intervention, outcomes and details of these (e.g., how measures), and key findings related to the scoping review question. The reviewers carefully saved the records to identify each study. Each reviewer charted each study because it may become apparent that additional unforeseen data can be usefully charted in this way.

## 9. Ethics and Dissemination

This paper does not require ethics approval as data are obtained through review of primary studies already published. This project will serve as a pilot for future studies at the University of Campania “Luigi Vanvitelli.” The results of our evaluation will be disseminated on author’s web sites. Additional dissemination will occur through presentations at conferences, such as courses and science education conferences, nationally and internationally, and through articles published in peer-reviewed journals. Finally, all team members will use their networks (i.e., Researchgate, LinkedIn and Twitter) to encourage dissemination of results. 

## Figures and Tables

**Table 1 mps-03-00016-t001:** Search strategy conducted on 7 February 2020.

Search	Query	Records Retrieved
PubMed	"antimicrobial lock therapy"[All Fields] OR "lock therapy"[All Fields] OR (("anti-infective agents"[Pharmacological Action] OR "anti-infective agents"[MeSH Terms] OR ("anti-infective"[All Fields] AND "agents"[All Fields]) OR "anti-infective agents"[All Fields] OR "antimicrobial"[All Fields]) AND ("lock therapy"[All Fields] OR "lock"[All Fields])) OR (("anti-infective agents"[Pharmacological Action] OR "anti-infective agents"[MeSH Terms] OR ("anti-infective"[All Fields] AND "agents"[All Fields]) OR "anti-infective agents"[All Fields] OR "antimicrobial"[All Fields]) AND lock[All Fields] AND ("therapy"[Subheading] OR "therapy"[All Fields] OR "therapeutics"[MeSH Terms] OR "therapeutics"[All Fields]))	952

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
