# Peer review of "Antimicrobial Lock Therapy in Clinical Practice: A Scoping Review Protocol"

_mps, 2020, doi:10.3390/mps3010016_

Round 1

Reviewer 1 Report

The authors present a methods paper on the development of a scoping review to map the evidence for antimicrobial lock therapy. This is an important and timely topic given the challenges associated with CLABSIs and the need to mitigate the growing threat of antimicrobial resistance. The scoping review appears to be well planned and organized. However, some additional information is required.  

BACKGROUND

Line 6 - Please add a reference to support statement that most BSIs are associated with intravascular devices

Line 17,18 - Surveillance and management strategies seem to pertain more to established BSI rather than catheter care. Consider revising this sentence. 

Line 33 - consider removing the first occurence of “cost”

Although scoping reviews tend to favour breadth over depth, I would suggest being more clear with the research question/objective in the background. I.e. is it to determine the optimal lock solution, whether ALT is effective and in which situations, etc?

FIGURE 1

Does not appear to add value to understanding the aim of the scoping review. Consider removing this figure. 

PROTOCOL DESIGN

Please provide further information about who will be involved (disciplines/expertise) in the scoping review and how public involvement will be achieved. 

Line 57 - it is indicated that substances other than antibiotics will be considered (e.g., citrate?). This should be clearly stated as included vs. excluded, since the title implies the focus is antimicrobials.

Is any limitation applied to the type of study design? How will observational studies and review articles be addressed?

Is grey literature out of scope?

Author Response

Response to Reviewer 1 Comments

The authors present a methods paper on the development of a scoping review to map the evidence for antimicrobial lock therapy. This is an important and timely topic given the challenges associated with CLABSIs and the need to mitigate the growing threat of antimicrobial resistance. The scoping review appears to be well planned and organized. However, some additional information is required.

BACKGROUND

Line 6 - Please add a reference to support statement that most BSIs are associated with intravascular devices

Response 1: Many thanks for the suggestion, we added a new reference as required in line 3 [New ref. Mayer J, Greene T, Howell J, et al. Agreement in classifying bloodstream infections among multiple reviewers conducting surveillance. Clin Infect Dis. 2012;55(3):364–370]

Line 17, 18 - Surveillance and management strategies seem to pertain more to established BSI rather than catheter care. Consider revising this sentence.

Response 2: We completely agree, we revised the sentence in: Antimicrobial lock therapy has been investigated in the management of central venous catheters, both for prevention and treatment of related infections (see lines 22-23)

Line 33 - consider removing the first occurrence of “cost”

Response 3: We removed the first occurrence of “cost” as suggested (see line 53)

Although scoping reviews tend to favour breadth over depth, I would suggest being more clear with the research question/objective in the background. I.e. is it to determine the optimal lock solution, whether ALT is effective and in which situations, etc?

Response 4: It is beyond our intentions use the results of our scoping review to determine the optimal lock solution, whether ALT is effective and in which situations, etc. It could be danger that the existence of studies rather than their intrinsic quality is used as the basis for generating conclusions. (see lines 75-79)

[New ref. Grant MJ, Booth A. A typology of reviews: an analysis of 14 review types and associated methodologies. Health Info Libr J. 2009;26(2):91–108.]

FIGURE 1

Does not appear to add value to understanding the aim of the scoping review. Consider removing this figure.

Response 5: We removed the figure as suggested

PROTOCOL DESIGN

Please provide further information about who will be involved (disciplines/expertise) in the scoping review and how public involvement will be achieved.

Response 5: We added in the Protocol design that “This scoping review began with the collaboration of a research team composed of experts in critical care medicine, infectious diseases, epidemiology and research synthesis”. Furthermore, we added how public involvement will be achieved (see lines 79-81)

Line 57 - it is indicated that substances other than antibiotics will be considered (e.g., citrate?). This should be clearly stated as included vs. excluded, since the title implies the focus is antimicrobials.

Response 6: However, catheter lock technique would be the most appropriate definition because it would include both antibiotic and non-antibiotic lock solutions in the current terminology ALT is used for all the techniques regardless of the lock solution used. We stated it in line 29-32

Is any limitation applied to the type of study design? How will observational studies and review articles be addressed?

Response 7: The observational studies that will have the criteria to be included will be discussed in the results (see stage 5). Otherwise, we will use the systematic reviews as a source for any observational studies not found in primary research (line 135)

Is grey literature out of scope?

Response 8: we completely agree with the reviewer, we added the grey literature (lines 128-134)

Reviewer 2 Report

Central catheter infections are a serious problem. Although antimicrobial lock therapy (ALT) is described extensively in literature, there is no consensus on the most appropriate ALT method. The authors propose a scoping review to answer 7 questions targeted at finding the patient group to benefit from ALT, as well as the ALT techniques and characteristics.

The manuscript contains many grammatical errors, and a thorough check of the entire manuscript is recommended.

ABSTRACT

The objective of the abstract seems to suggest that the article itself is a review, however the main body describes a scoping review protocol. The abstract and main text must provide a clear objective, rationale and steps to complete the objective, which is currently not the case.

 “setting” should be “settings”.

Key question 2 seems to be the same question as question 7.

The inclusion criteria in the abstract are not the same as those in line 65-68 of the main body. Please be consistent.

Please capitalize “and” in last sentence.

Please use capital letters at the beginning of sentences or titles.

MAIN BODY

It is unclear what the proposed scoping review will add to what we already know from existing reviews on ALT (please have a look at Vassallo et al. (2015) Antimicrobial lock therapy in central-line associated bloodstream infections: a systematic review).

Line 5-6: please provide a reference to support this claim.

Line 6: Please use plural “CLABSIs” instead of singular “CLABSI”

Line 9: please provide a reference to support this claim.

Line 12: Why is the worldwide CLABSI rate 4.1 /1000 days compared to 0.8 in the U.S? Also, provide a literature reference.

Line 13: Please specify which “other” healthcare-associated infections are meant.

Line 15-16: should be in a single same paragraph with lines 17-20.

Line 21: Please explain the basic principles of ALT

Line 25-26 “All these factors delayed the inclusion of ALT into guidelines for the management of CLABSI until 2009”. Please elaborate how the application ALT is recommended in the 2009 guidelines.

Line 28 “strictly recommended” is a contradiction. Do you mean “highly recommended” or “strictly imposed”?

Figure 1. I find this figure to be quite confusing. It is unclear whether differences in the size or color of the text boxes have any meaning. Do they reflect the relative importance of each characteristic?  In its current form Figure 1 does add any new information to the list in lines 32-35.

Line 36. This is the first time you mention a “scoping review”, however it is written as if the reader should already be familiar with the objective of the protocol. Please provide a proper introduction to the reader.

Line 36-37. The previous paragraph suggests that the main issue is that the most appropriate solution has not been determined. Please summarize the reasons that justify the objective of your scoping review.

How are patients and public involved in this study? It is also unclear why and if they should be involved in the first place.

49-51. It is not clear what is meant in this sentence. Please rewrite in shorter sentences and provide clear examples of what you mean.

53-63. The contents of this paragraph should be presented in a more structured manner. Consider presenting the data in a table.

In the abstract it is mentioned that the included studies all must be controlled studies, however it is not mentioned here.

85-87. “Two authors (AA and SDF) independently will evaluate the titles and abstracts of potentially eligible studies.” What happens when there is no consensus on the eligibility of a study?

Please specify what is meant by “JBI”. “The reviewers will keep careful records to identify each study.” Could you be more specific on the data that is logged? Change “founded” to “funded”

REFERENCES

Line 183, 185 (ref 15 & 16) add spacing before publication year.

Line 193-194 Ref 19. It seems a publication year is missing.

Author Response

Response to Reviewer 2 Comments

Central catheter infections are a serious problem. Although antimicrobial lock therapy (ALT) is described extensively in literature, there is no consensus on the most appropriate ALT method. The authors propose a scoping review to answer 7 questions targeted at finding the patient group to benefit from ALT, as well as the ALT techniques and characteristics.

The manuscript contains many grammatical errors, and a thorough check of the entire manuscript is recommended.

Response 1: Many thanks for the suggestion, Dr. Nyias polished the language of the entire manuscript

ABSTRACT

The objective of the abstract seems to suggest that the article itself is a review, however the main body describes a scoping review protocol. The abstract and main text must provide a clear objective, rationale and steps to complete the objective, which is currently not the case.

Response 2: Many thanks for the suggestion, we changed the abstract

OBJECTIVE: Our objective is to review the scientific literature on the use of Antimicrobial Lock Therapy (ALT). To achieve this result, our scoping review will address the following seven key questions:

Who are the patients who will benefit from this technique? What are the techniques utilized? What are the settings in which the technique is performed? When the technique is performed? Why the technique is performed? How the technique is performed? In how much amount, of such technique performed?

INTRODUCTION: Central venous catheter infections are becoming a very serious problem due to increasingly intensive and invasive treatments, with an enormous economic impact. The Infectious Diseases Society of America recommends ALT as adjunctive therapy specifically for catheter salvage in cases where the catheter is not removed. Despite these considerations, the most appropriate lock therapy for central venous access devices is still to be defined.

INCLUSION CRITERIA: This review will consider all studies, excluding experimental models, focusing on the ALT. Only studies published in full and in peer-reviewed journals will be included, with no restrictions on language, on the year of publication and age of the participants. Both randomized controlled trials and observational studies (including cohort and case-control controlled and uncontrolled studies) will be included.

METHODS: This scoping review has been planned on the five stages framework: 1. Identifying the review question; 2. Identifying relevant studies; 3. Study selection; 4. Charting the data; 5. Collating, summarizing and reporting the results. It will be conducted in accordance with the Preferred Reporting Items for Systematic Reviews and Meta-Analyses (PRISMA-ScR) Guidelines. The databases utilized will include MEDLINE via PubMed, Embase and Cochrane Central Register of Controlled Trials and Grey Literature.

SCOPING REVIEW REGISTRATION: Open Science Framework https://osf.io/vphwm/

“setting” should be “settings”.

Response 3: We corrected it.

Key question 2 seems to be the same question as question 7.

Response 4: Many thanks, we changed it.

The inclusion criteria in the abstract are not the same as those in line 65-68 of the main body. Please be consistent.

Response 5: We apologize and corrected it.

Please capitalize “and” in last sentence.

Response 6: We apologize and corrected it.

Please use capital letters at the beginning of sentences or titles.

Response 7: We apologize and corrected it.

MAIN BODY

It is unclear what the proposed scoping review will add to what we already know from existing reviews on ALT (please have a look at Vassallo et al. (2015) Antimicrobial lock therapy in central-line associated bloodstream infections: a systematic review).

Response 8: Many thanks for the opportunity to clarify the purpose of our study “this scoping review unlike the previous systematic reviews (Vassallo et al. 2015) does not aim to give recommendations on the use of ALT but aims to furnish a map and a synthesis on this topic, offering a base for the development of new clinical practice indications and health policies” (Line 63-67)

Line 5-6: please provide a reference to support this claim.

Response 1: We apologize for the missing, we have added it (lines 5-6)

Line 6: Please use plural “CLABSIs” instead of singular “CLABSI”

Response 2: We apologize for the imprecision, we corrected it (line 9)

Line 9: please provide a reference to support this claim.

Response 3: Thanks, we have added it (line 9 reference 3.)

Line 12: Why is the worldwide CLABSI rate 4.1/1000 days compared to 0.8 in the U.S? Also, provide a literature reference.

Response 4: Thanks, we have added it (lines 10-12)

Line 13: Please specify which “other” healthcare-associated infections are meant.

Response 5: We have added a description in parenthesis (lines 13-15)

Line 15-16: should be in a single same paragraph with lines 17-20.

Response 6: We corrected it (lines 16-21)

Line 21: Please explain the basic principles of ALT

Response 7: We have added a short description of the principles of the ALT (lines 23-29 and reference number 9)

Line 25-26 “All these factors delayed the inclusion of ALT into guidelines for the management of CLABSI until 2009”. Please elaborate how the application ALT is recommended in the 2009 guidelines.

Response 8: We have added a short description of what was recommended in the 2009 guidelines (lines 37-40)

Line 28 “strictly recommended” is a contradiction. Do you mean “highly recommended” or “strictly imposed”?

Response 9: We meant highly recommended, we corrected it (line 42)

Figure 1. I find this figure to be quite confusing. It is unclear whether differences in the size or color of the text boxes have any meaning. Do they reflect the relative importance of each characteristic? In its current form Figure 1 does add any new information to the list in lines 32-35.

Response 10: We have removed Figure 1 as suggested by Reviewer 1.

Line 36. This is the first time you mention a “scoping review”, however it is written as if the reader should already be familiar with the objective of the protocol. Please provide a proper introduction to the reader.

Response 11: We provided a short introduction on the scoping review (lines 75-79)

Line 36-37. The previous paragraph suggests that the main issue is that the most appropriate solution has not been determined. Please summarize the reasons that justify the objective of your scoping review.

Response 12: we summarized the aims and the reason of our scoping review (lines 67-70)

How are patients and public involved in this study? It is also unclear why and if they should be involved in the first place.

Response 13: In this scoping review no patients nor public is involved directly. However, the authors have the responsibility to furnish data that can be used to build new clinical practice indications and to guide health policies influencing patients and public treatments and outcomes. So that, we have planned to involve associations of patients, scientific societies and health care professionals.

49-51. It is not clear what is meant in this sentence. Please rewrite in shorter sentences and provide clear examples of what you mean.

Response 13bis: We apologize, we corrected it (line 99)

53-63. The contents of this paragraph should be presented in a more structured manner. Consider presenting the data in a table.

Response 14: We apologize, we corrected it (lines 100-109)

In the abstract it is mentioned that the included studies all must be controlled studies, however it is not mentioned here.

Response 15: We apologize for the inaccuracy, due to the nature of our study we decided to include randomized and non-randomized studies, including uncontrolled studies (lines 112-113)

85-87. “Two authors (AA and SDF) independently will evaluate the titles and abstracts of potentially eligible studies.” What happens when there is no consensus on the eligibility of a study?

Response 16: We apologize for the missing, we have added that any disagreements on study eligibility between AA and SDF will be resolved according to a discussion with a third reviewer (lines 139-141).

Please specify what is meant by “JBI”. “The reviewers will keep careful records to identify each study.” Could you be more specific on the data that is logged? Change “founded” to “funded”

Response 17: In Stage 4 we specify what JBI mean (line 145). We changed as requested “founded” to “funded”

REFERENCES

Line 183, 185 (ref 15 & 16) add spacing before publication year.

Response 18: We apologize for the error; we have added it.

Line 193-194 Ref 19. It seems a publication year is missing.

Response 19: We apologize for the error; we have added it.

Reviewer 3 Report

Interesting subject, important potential results.

Please re-check the document for typo errors (e.g. `managment`).

Some (more) articles could be important to use in the `background` as well for the entire document, e.g.

Norris LB, et al. Systematic review of antimicrobial lock therapy for prevention of central-line-associated bloodstream infections in adult and pediatric cancer patients. Int J Antimicrob Agents. 2017;50(3):308-317. doi: 10.1016/j.ijantimicag.2017.06.013.

Zembles TN,et al. Development and implementation of an antimicrobial lock therapy guideline in a pediatric hospital. Am J Health Syst Pharm. 2018;75(5):299-303. doi: 10.2146/ajhp161056.

Arechabala MC, et al. Antimicrobial lock solutions for preventing catheter-related infections in haemodialysis. Cochrane Database Syst Rev. 2018;4:CD010597. doi: 10.1002/14651858.CD010597.pub2.

Dang FP, et al. Comparative efficacy of various antimicrobial lock solutions for preventing catheter-related bloodstream infections: A network meta-analysis of 9099 patients from 52 randomized controlled trials. Int J Infect Dis. 2019;87:154-165. doi: 10.1016/j.ijid.2019.08.017.

Basas J, et al. Efficacy of liposomal amphotericin B and anidulafungin using an antifungal lock technique (ALT) for catheter-related Candida albicans and Candida glabrata infections in an experimental model. PLoS One. 2019;14(2):e0212426. doi: 10.1371/journal.pone.0212426.

The figure is original, modified, other? Please explain.

Line 50: please explain about stakeholders (more concrete).

Eligibility. English language could be restrictive. It is possibile to find good articles, good ideas in articles written in other languages. This restriction looks not appropriate (even the work would be more extensive).

Search in MEDLINE and CINAHL only is restrictive. Should be enlarged.

More is to be explained / extended in the `Dissemination` part of the document. E.g. regargind what team members (who, how many ?) will do for `broad dissemination`.

Author Response

Response to Reviewer 3 Comments

Comments and Suggestions for Authors

Interesting subject, important potential results.

Please re-check the document for typo errors (e.g. `managment`).

Response 1: We apologize for the error; we corrected it

Some (more) articles could be important to use in the `background` as well for the entire document, e.g.

Norris LB, et al. Systematic review of antimicrobial lock therapy for prevention of central-line-associated bloodstream infections in adult and pediatric cancer patients. Int J Antimicrob Agents. 2017;50(3):308-317. doi: 10.1016/j.ijantimicag.2017.06.013.

Zembles TN,et al. Development and implementation of an antimicrobial lock therapy guideline in a pediatric hospital. Am J Health Syst Pharm. 2018;75(5):299-303. doi: 10.2146/ajhp161056.

Arechabala MC, et al. Antimicrobial lock solutions for preventing catheter-related infections in haemodialysis. Cochrane Database Syst Rev. 2018;4:CD010597. doi: 10.1002/14651858.CD010597.pub2.

Dang FP, et al. Comparative efficacy of various antimicrobial lock solutions for preventing catheter-related bloodstream infections: A network meta-analysis of 9099 patients from 52 randomized controlled trials. Int J Infect Dis. 2019;87:154-165. doi: 10.1016/j.ijid.2019.08.017.

Basas J, et al. Efficacy of liposomal amphotericin B and anidulafungin using an antifungal lock technique (ALT) for catheter-related Candida albicans and Candida glabrata infections in an experimental model. PLoS One. 2019;14(2):e0212426. doi: 10.1371/journal.pone.0212426.

Response 2: We thank the reviewer for the suggestion, we have extended the background with the suggested references.

The figure is original, modified, other? Please explain.

Response 3: The figure was original, but it has been removed as suggested by the Reviewer 1.

Line 50: please explain about stakeholders (more concrete).

Response 4: We have added in the text the description of what we meant for stakeholders in parenthesis (lines 91-92).

Eligibility. English language could be restrictive. It is possible to find good articles, good ideas in articles written in other languages. This restriction looks not appropriate (even the work would be more extensive).

Response 5: We agree with the suggestion and we will enlarge the search without language restrictions (line 113).

Search in MEDLINE and CINAHL only is restrictive. Should be enlarged.

Response 6: We completely agree the suggestion, MEDLINE and CINAHL were the database for the primary step, in the secondary step will be used MEDLINE, EMBASE and CENTRAL (lines 122-123)

More is to be explained / extended in the `Dissemination` part of the document. E.g. regargind what team members (who, how many ?) will do for `broad dissemination`

Response 7: Many thanks, we specified that all team members will use their networks (i.e. Researchgate, LinkedIn and Twitter) to encourage dissemination of results (line 184)

Round 2

Reviewer 1 Report

Thank you for these changes, they improve the clarity of the objectives for this work. The revised manuscript is satisfactory.